# Exaggerated Autophagy in Stanford Type A Aortic Dissection: A Transcriptome Pilot Analysis of Human Ascending Aortic Tissues

**DOI:** 10.3390/genes11101187

**Published:** 2020-10-13

**Authors:** Zeyi Zhou, Yan Liu, Xiyu Zhu, Xinlong Tang, Yali Wang, Junxia Wang, Can Xu, Dongjin Wang, Jie Du, Qing Zhou

**Affiliations:** 1Department of Thoracic and Cardiovascular Surgery, Affiliated Drum Tower Hospital of Nanjing University Medical School, Institute of Cardiothoracic Vascular Disease, Nanjing University, Nanjing 210008, China; dg1735106@smail.nju.edu.cn (Z.Z.); dg1635108@smail.nju.edu.cn (X.Z.); jsxinlongtang@njglyy.com (X.T.); mg1835042@smail.nju.edu.cn (Y.W.); dg1935107@smail.nju.edu.cn (J.W.); skytiankong1023@smail.nju.edu.cn (C.X.); wangdongjin@njglyy.com (D.W.); 2Beijing Anzhen Hospital, Capital Medical University, Beijing Institute of Heart Lung and Blood Vessel Diseases, Beijing 100029, China; yliu@bjmu.edu.cn; 3The Key Laboratory of Remodeling-Related Cardiovascular Diseases, Ministry of Education, Beijing 100029, China

**Keywords:** Stanford type A aortic dissection, molecular pathology, exaggerated autophagy, transcriptome sequencing, human ascending aortic tissues

## Abstract

Stanford type A aortic dissection (TAAD) is one of the most dangerous diseases of acute aortic syndrome. Molecular pathological studies on TAAD can aid in understanding the disease comprehensively and can provide insights into new diagnostic markers and potential therapeutic targets. In this study, we defined the molecular pathology of TAAD by performing transcriptome sequencing of human ascending aortic tissues. Pathway analysis revealed that activated inflammation, cell death and smooth muscle cell degeneration are the main pathological changes in aortic dissection. However, autophagy is considered to be one of the most important biological processes, regulating inflammatory reactions and degenerative changes. Therefore, we focused on the pathological role of autophagy in aortic dissection and identified 10 autophagy-regulated hub genes, which are all upregulated in TAAD. These results indicate that exaggerated autophagy participates in the pathological process of aortic dissection and may provide new insight for further basic research on TAAD.

## 1. Introduction

Acute aortic dissection is a sudden onset of blood leaving the normal aortic lumen through an intimal tear and rapidly dissecting the inner layers from the outer layers of the media to produce a false lumen [1,2]. Stanford type A aortic dissection, also known as type A aortic dissection (TAAD), is one of the most common dissections [3]. The treatment of TAAD involves surgical and hybrid endovascular techniques; however, the mortality and morbidity rates remain high [4].

With the continuous advancement of gene sequencing technology, molecular pathology has gradually become an independent discipline of science. This technology provides better diagnostic indicators and facilitates the development of treatment methods for many diseases [5,6], especially cancers. Previous studies on aortic dissection have always used animal models, which cannot aid in fully understanding its pathogenesis owing to various limitations. Hence, we used human ascending aortic tissues for our study. We performed whole transcriptome sequencing to acquire differential gene expression profiles in 10 TAAD tissues and 10 normal tissues.

Autophagy is a biological process in which lysosomes degrade damaged organelles and macromolecular substances, thereby playing an important role in stabilizing the intracellular environment [7]. It has been demonstrated that exaggerated autophagy can be triggered by the disruption of cellular homeostasis in various kinds of cells and is highly related to diseases such as cancer, cardiovascular diseases and Alzheimer’s disease [8,9,10,11]. To detect whether autophagy contributes to the development of TAAD [12], we identified autophagy-related genes (ARGs) from the Autophagy Database and the Human Autophagy Database. By comparing differentially expressed genes (DEGs) obtained from the sequencing results for ARGs, we identified the differentially expressed ARGs (DEARGs) in TAAD. Gene ontology (GO) and Kyoto Encyclopedia of Genes and Genomes (KEGG) analyses of DEARGs and protein–protein interaction (PPI) network analyses using Cytoscape software were used to determine the key modules and hub genes.

## 2. Materials and Methods

### 2.1. Ascending Aortic Tissue Sample Collection and Transcriptome Sequencing

This study was conducted in accordance with the Declaration of Helsinki and was approved by the Medical Ethics Committee of Nanjing Drum Tower Hospital, the affiliated hospital of Nanjing University Medical School (Institutional Review Board File 2016-152-01). The ascending aortic tissue samples used in this study were obtained from the Department of Cardiothoracic Surgery of Nanjing Drum Tower Hospital in 2017. All patients with TAAD were homogenous and none had hereditary diseases, such as Marfan syndrome, Loeys–Dietz syndrome and bicuspid aortic valve malformation. We measured the inner diameter of the ascending aorta, where the largest dilation occurred, and we considered this as the aortic diameter, based on the results of the preoperative computed tomography angiography (CTA) examination. We did not detect the inner diameter of normal samples because routine inspections did not report any abnormalities. All patients with TAAD underwent ascending aortic replacement surgery during a cardiopulmonary bypass. Normal ascending aortic tissue samples were obtained from patients undergoing coronary artery bypass grafting surgery (CABG) without any aortic diseases. We selected 20 samples (10 TAAD and 10 normal) for whole transcriptome sequencing. The TAAD ascending aorta sample was cut above the sinutubular junction and completely transected just proximal to the origin of the brachiocephalic artery, whereas the normal ascending aorta sample was collected by aortic punch during CABG. Total RNA was extracted from each sample using TRIzol Reagent (Thermo Fisher, Carlsbad, CA, USA). Subsequent sequencing was completed by Novel-Bioinformatics Ltd. Co. (Shanghai, China). RNA library construction was performed using standard Illumina protocols. The library products were then sequenced on Illumina HiSeq X Ten, and generated no less than 10 M paired-end reads for each sample. Raw reads were trimmed for adapter sequences, masked for low-complexity or low-quality sequences and mapped to the hg19 whole genome using Tophat2 (version 2.0.9). Raw data and processed data are available at the Gene Expression Omnibus (GEO accession: GSE153434).

### 2.2. Identification of DEGs

DEGs between the TAAD and normal samples were identified using the DESeq2 package (version 3.11.0) in R software (version 3.6.1). The Benjamini and Hochberg-corrected *p* value of < 0.05 and |LogFC| > 1 were defined as the selection thresholds for selecting the DEGs.

### 2.3. GO and KEGG Enrichment Analyses

We used the clusterProfiler package (version 3.12.0) in R to perform the GO and KEGG enrichment analyses of DEGs (*p* < 0.05).

### 2.4. Identification of DEARGs

Seven hundred and forty-three ARGs were identified from the Autophagy Database (http://www.tanpaku.org/autophagy/list/GeneList.html) and another 231 ARGs were identified from the Human Autophagy Database (http://www.autophagy.lu/clustering/). After deduplication of genes from these two databases, we obtained 847 ARGs. Using the VennDiagram package (1.6.20) in R, we selected 74 DEARGs.

### 2.5. PPI Network and Modular Analyses

A PPI network of DEARGs was constructed using the STRING database (version 11.0) and visualization was performed using Cytoscape software (version 3.8.0). The Cytoscape plugin MCODE was applied to identify four functional modules containing 31 DEARGs (the parameters were set to default: degree cutoff = 2, node score cutoff = 0.2, K-core = 2 and Max depth = 100). Another plugin, Cytohubba, was used to detect hub genes. The built-in algorithm of Cytohubba assigned a value to each gene in the PPI network and sorted these genes by values. Genes with a value larger than 10 were meaningful and regarded as hub genes.

### 2.6. Hub Gene Validation

The online GSE52093 dataset, obtained from the Gene Expression Omnibus database, was used for validation. This dataset contains seven TAAD aortic tissue samples and five normal aortic tissue samples. Raw data of this dataset were downloaded and normalized by quantile normalization using IlluminaGUI in R. Receiver operating characteristic curves were made using GraphPad Prism (version 8.0). Four normal and 7 TAAD ascending aorta samples which were not used for transcriptome sequencing were selected for validation of hub gene expression. Clinical information on these 11 samples is presented in Appendix A. Total RNA was extracted from each sample using TRIzol Reagent (Thermo Fisher). First-strand cDNA synthesis was performed using a First Strand cDNA Synthesis Kit (Life Technologies, Carlsbad, CA, USA). TB Green Premix ex Taq II (Takara, Dalian, China) was used for quantitative PCR (qPCR) in a CFX connected Real-time System (Bio-Rad, Hercules, CA, USA). We used β-Actin as a control. PCR primers were described in Appendix A.

## 3. Results

### 3.1. Overall Protocol of the Study

The flowchart of our study is shown in Figure 1 and the clinical information for the TAAD and normal samples is given in Table 1. Noncoding RNA analysis of the dataset has already been published [13].

### 3.2. Identification of DEGs

A total of 1525 DEGs were identified from our datasets, of which 542 were upregulated and 983 were downregulated (Figure 2A). The heatmap of DEG expression revealed that mRNA expression between the TAAD and normal samples was distinct (Figure 2B). DEGs were listed in Appendix A.

### 3.3. GO and KEGG Enrichment

To understand the functions and related pathways of the DEGs, we performed GO functional and KEGG enrichment analyses. GO functional analysis was divided into three components: “biological process” (BP), “cellular component” (CC), and “molecular function” (MF). In the BP component, upregulated and downregulated genes were mainly enriched for leukocyte migration and muscle tissue development, respectively. Furthermore, in CC, the upregulated genes were mainly enriched for platelet alpha granules and extracellular matrix, whereas the downregulated genes were enriched for the extracellular matrix. In the MF component, the upregulated genes were mainly enriched for platelet-derived growth factor receptor binding and amino acid transmembrane transporter activity, whereas the downregulated genes were enriched for the extracellular matrix structural constituent and glycosaminoglycan binding pathways (Figure 3A,B). The KEGG pathway enrichment analysis showed that the upregulated genes were enriched for the tumor necrosis factor signaling pathway, interleukin-17 signaling pathway, the cytokine–cytokine receptor interactions, ferroptosis and phosphatidylinositol 3-kinase/Akt pathway, whereas the downregulated genes were enriched for the Wnt signaling pathway, axon guidance, circadian entrainment and vascular smooth muscle contraction processes. Many of these pathways are associated with cell death and growth, which contribute to smooth muscle cell degeneration.

### 3.4. Identification of DEARGs

Autophagy plays an important role in intracellular homeostasis. To evaluate the activation of autophagy in TAAD, we obtained ARGs from the Autophagy Database and the Human Autophagy Database. After deduplication of genes from these two databases, we obtained 847 ARGs (Appendix A). We compared these with the DEGs in TAAD and selected 74 DEARGs (Figure 4A, Appendix A). Furthermore, we performed GO functional analysis and KEGG enrichment analysis on the DEARGs. These genes were enriched for protein tyrosine kinase activity, ubiquitin protein ligase binding and protein serine/threonine kinase activity from the molecular functional component of the GO analysis (Figure 4B,C). These results indicate that protein modifications play an important role in the pathogenesis of TAAD.

### 3.5. PPI Network and Its Modular Analysis

A PPI network of DEARGs was constructed with 74 gene nodes and 243 edges using the STRING database (Figure 5A, Appendix A). Further, we used Cytoscape software to analyze the data. The Cytoscape plugin MCODE displayed four functional modules containing 31 DEARGs (Figure 5B, Appendix A). The results of GO and KEGG enrichment analyses of genes in the four modules were similar to those of the enrichment analysis of all DEARGs (Figure 5C,D). This could mean that these 31 genes are the core genes of autophagy in TAAD. Crosstalk analysis of gene function and pathways among the four modules revealed that module one is closely connected to module two (Figure 5E,F, Appendix A). Furthermore, we used the Cytohubba plugin to detect hub genes (Figure 5G). We sorted these genes by values and selected genes with values larger than 10 as hub genes. Hence, we selected *BDNG*, *CCL2*, *CDKN1A*, *FOS*, GAPDH, *HIF1A*, *JUN*, *MYC*, *SOD2* and *VEGFA* as hub genes.

### 3.6. Validation of Hub Genes in GSE52093

We validated the expression of hub genes in GSE52093. Seven of the 10 hub genes (*BDNG*, *CCL2*, *CDKN1A*, *HIF1A*, *MYC*, *SOD2* and *VEGFA*) were significantly upregulated in the aortic dissection samples compared with the normal samples (Figure 6A), consistent with our analysis. We also used qPCR to detect the hub genes’ expression in a new cohort which contained four normal and seven TAAD ascending aorta samples. Clinical information is presented in Appendix A. Results of qPCR showed that SOD2, HIF1A and VEGFA were significantly upregulated in TAAD samples (Appendix A). Analysis of receiver operating characteristic (ROC) curves suggested that these genes may be potential biomarkers for TAAD diagnosis (Figure 6B). This may be especially true for *HIF1A* (area under the ROC curve, 1.000). Proteins encoded by the autophagy related (ATG) gene family (*ULK1*, *ATG2A*, *ATG2B*, *ATG3*, *ATG4A*, *ATG4B*, *ATG4C*, *ATG4D*, *ATG5*, *ATG7*, *ATG9A*, *ATG9B*, *ATG10*, *ATG12*, *ATG13*, *ATG14*, *ATG16L1* and *ATG16L2*) are considered the key proteins that can directly regulate the process of autophagy [14,15]. To determine which protein in the ATG family is highly related to *HIF1A*, we performed a correlation analysis between *HIF1A* and the ATG family proteins from our dataset and the GSE52093 dataset. The results showed that *ATG3* was related to *HIF1A* in both datasets and the correlation between *HIF1A* and *ATG3* was positive (Figure 6C). *HIF1A* is recognized as the first adapter response to hypoxia, which can transmit signals and trigger a series of physiological responses [16]. This indicates that the *HIF1A-ATG3* axis may be the key factor in TAAD pathogenesis.

## 4. Discussion

Various animal models have been established to explore the molecular pathology of TAAD and each has its limitations. For example, apolipoprotein E-deficient mice (Apoe^−/−^ mice) infused with angiotensin II can only form abdominal aortic aneurysm or dissection [17]. Furthermore, β-aminopropionitrile fumarate (BAPN), which is known to inhibit lysyl oxidase, is one of the most common drugs for establishing a mouse model of thoracic aortic dissection [18,19]. However, mice fed with BAPN barely have hypertension, which is the most common symptom in patients with TAAD. As a systemic drug, BAPN can cause not only degradation of the extracellular matrix of aortic smooth muscles, but also degradation of cross-linked collagen in other tissues and organs, which can induce additional diseases such as uterine prolapse [20]. Therefore, we chose to perform experiments on human tissues in our study. Considering that there are few transcriptomics studies on aortic dissection, we performed whole transcriptome sequencing of TAAD tissue samples and compared them with sequencing results obtained from the tissues of healthy individuals.

Over the past 10 years, the rapid progress in various sequencing technologies has improved our understanding of diseases from simple morphological and histopathological studies to complicated molecular pathological studies [5]. Through molecular pathological studies, we have obtained new tumor diagnostic markers, such as *HER-2*, *EGFR* and *KRAS*. Furthermore, many therapeutic drugs specific to various cancers, such as trastuzumab, gefitinib and rituximab, have also been identified [21,22]. However, no study on aortic dissection has been able to completely reveal its molecular pathology. Hence, our knowledge of aortic dissection is limited to clinical manifestations, radiography and histopathology [12,23]. The diagnosis and treatment strategies based on these aspects are unsatisfactory, and the rates of misdiagnosis and mortality remain high [24]. Therefore, molecular pathological studies on aortic dissection are necessary.

In this study, we identified exaggerated autophagy as a molecular characterization of TAAD. More studies have begun to focus on smooth muscle cell (SMC) autophagy in aortic dissection [25,26,27,28]. However, the relationship between autophagy and aortic dissection remains controversial. On the one hand, excessive autophagic activation may induce SMC death and phenotype switch in aortic SMCs [27,29]. On the other hand, appropriate activation of autophagy may have a therapeutic effect on aortic aneurysm or dissection [30,31]. As previous studies were based on animal models, we propose that an analysis of results obtained from human tissue samples would be more reliable.

Genes identified from the Autophagy Database and the Human Autophagy Database represent a relatively comprehensive set of ARGs. Enrichment analysis of DEARGs revealed that post-transcriptional modifications of proteins such as phosphorylation and ubiquitination are key to the regulation of autophagy in aortic dissection (Figure 4B,C). It has been demonstrated that ubiquitination is an important and central regulation mechanism in the process of autophagy [32,33,34]. Disturbances to this mechanism can induce various diseases, such as inflammatory bowel disease, cardiac remodeling and cancer [35,36,37,38]. Furthermore, one study reported that a SMAD4 heterozygous variant that can cause increased SMAD4 ubiquitination is associated with thoracic aortic dissection in a family [39]. Therefore, we hypothesized that post-transcriptional modifications caused by autophagy, especially ubiquitination, play an important role in aortic dissection.

*HIF1A*, known as hypoxia-inducible factor 1α, serves as a regulator of the adaptive response to hypoxia [16,40]. Autophagy induced by hypoxia is a key factor in the progression of various tumors [41,42,43]. Although *HIF1A* activation likely induces the development of aortic dissection, the relationship between autophagy and hypoxia is poorly understood [44,45]. Autophagosomes are double-membrane vesicles formed during autophagy to engulf intracellular material and deliver cargo to lysosomes for their degradation or recycling into the vacuole [14,46]. Autophagy-related (ATG) proteins are considered key molecules in the biogenesis of autophagosomes [15,47]. The correlation analysis between *HIF1A* and the ATG family genes predicted that *HIF1A* has a tight connection with the *ATG3* gene in aortic dissection (Figure 6C). Although the R values of the correlation analysis in the two datasets are not high, this may be because *HIF1A* and *ATG3* are not directly related to each other. We speculate that the *HIF1A-ATG3* axis is an important factor in the pathogenesis of TAAD, and it is necessary to address the role of *ATG3* in aortic dissection.

Our study also has limitations. Owing to the urgency to provide treatment to patients with TAAD and its surgical requirements, basic and follow-up patient data were incomplete. This makes it impossible to perform further analysis on molecular features with specific clinical symptoms and postoperative prognosis. Meanwhile, results of this study could not sufficiently illustrate that the aortic dissection is caused by exaggerated autophagy. In order to prove this hypothesis, animal models are still needed for further verification. However, we believe that with the development of technology and further research on the molecular pathology of TAAD, a complete understanding of the aortic dissection can be obtained.

In conclusion, we identified differentially expressed genes in the normal and dissected ascending aorta that may be responsible for structural abnormalities predisposing patients to TAAD. We also identified 10 hub genes and the *HIF1A-ATG3* axis involved, which could provide new insights into the understanding of aortic dissection.

## Figures and Tables

**Figure 1 genes-11-01187-f001:**
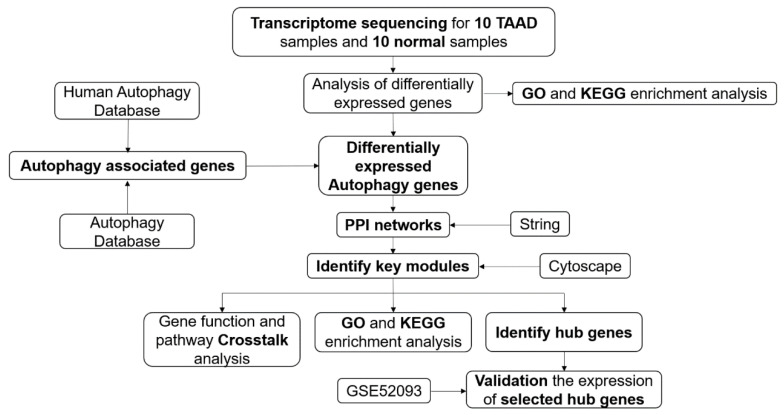
Overall protocol of the study. TAAD, type A aortic dissection; PPI, protein–protein interaction.

**Figure 2 genes-11-01187-f002:**
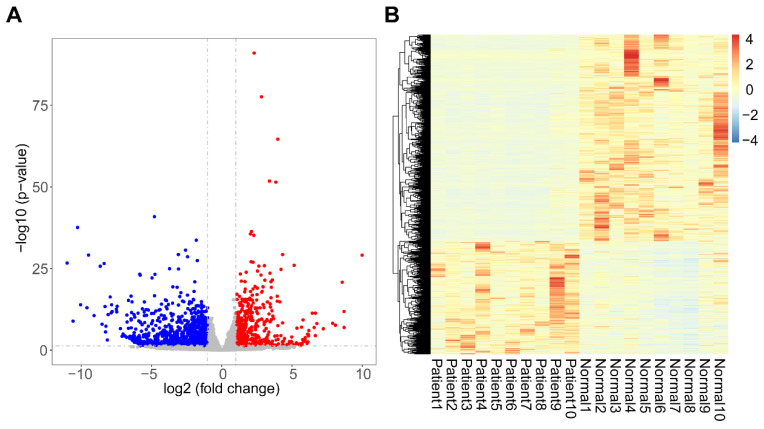
Identification of differentially expressed genes (DEGs). (**A**) Volcano plot of DEGs; red indicates up-regulated genes; blue indicates down-regulated genes. (**B**) Heatmap showing the expression of DEGs.

**Figure 3 genes-11-01187-f003:**
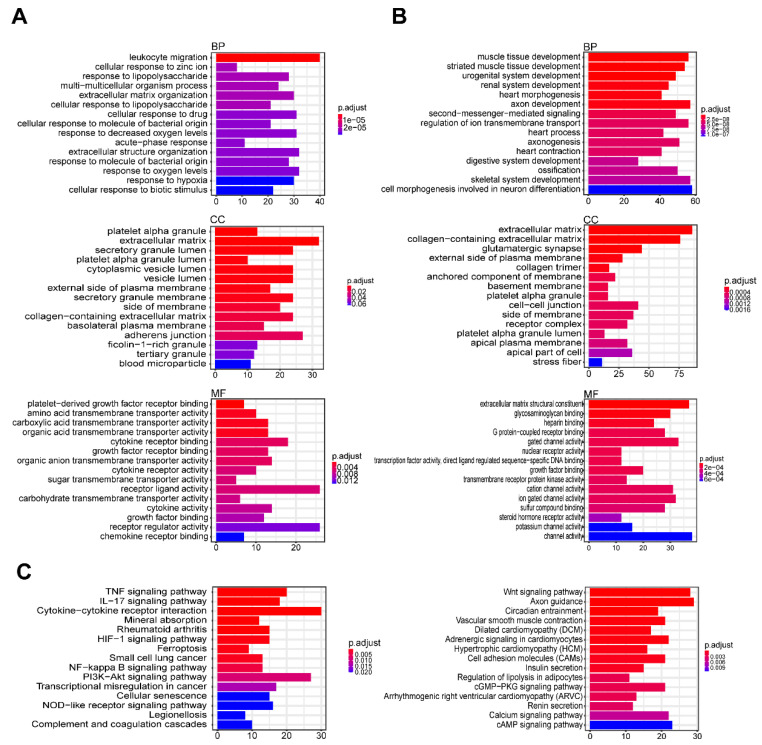
Gene ontology (GO) and Kyoto Encyclopedia of Genes and Genomes (KEGG) enrichment analysis. (**A**) GO enrichment analysis of upregulated genes. (**B**) GO enrichment analysis of downregulated genes. (**C**) KEGG pathway enrichment analysis of DEGs, left showing KEGG pathway of upregulated genes; right showing KEGG pathway of downregulated genes.

**Figure 4 genes-11-01187-f004:**
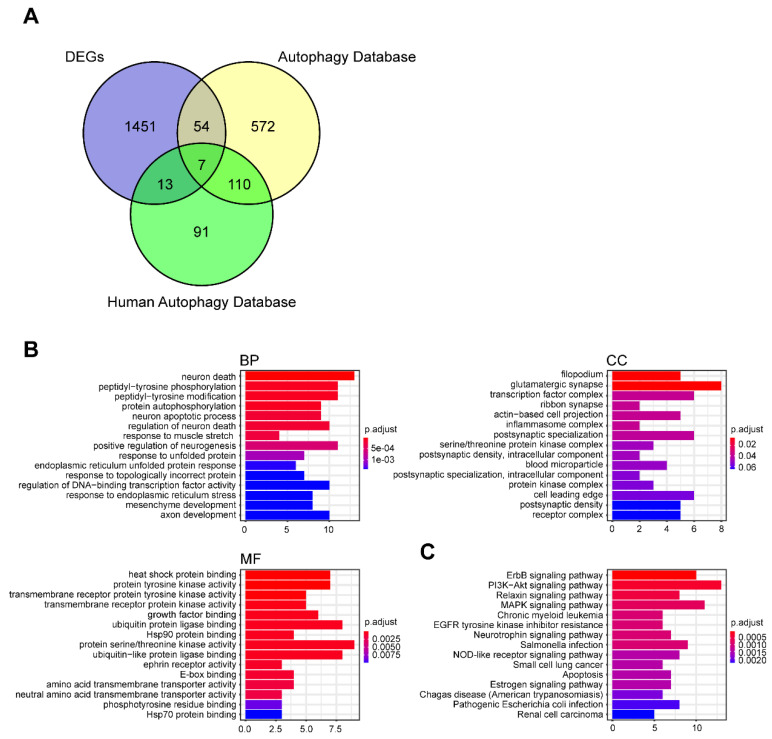
Identification of differentially expressed autophagy-related genes. (**A**) Venn diagram showing the overlap of genes between TAAD DEGs and autophagy associated genes. (**B**) GO enrichment analysis of overlap genes. (**C**) KEGG pathway enrichment analysis of overlap genes.

**Figure 5 genes-11-01187-f005:**
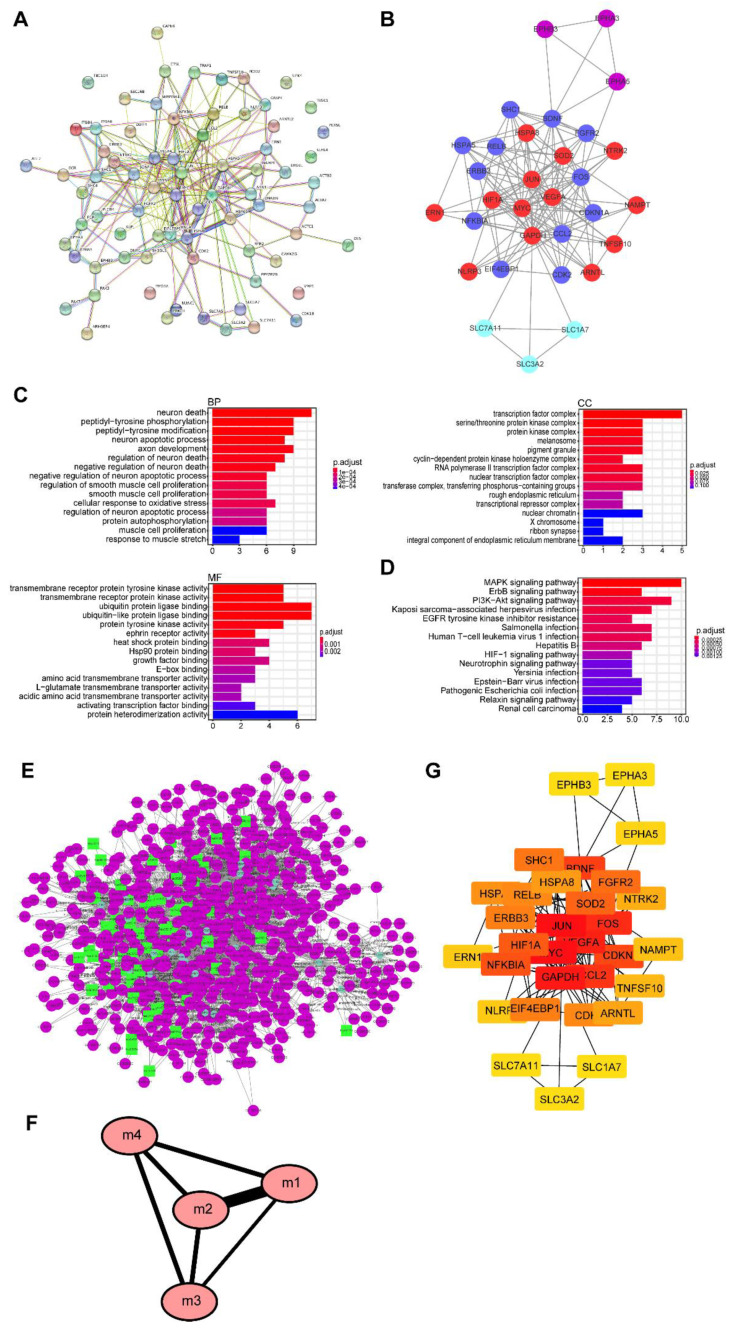
PPI network and its modular analysis. BP: biological process; CC: cell component; MF: molecular function. (**A**) PPI network. (**B**) Gene modules and related genes; blue: module one; red: module two; cyan-blue: module three; purple: module four. (**C**) GO enrichment analysis of modules. (**D**) KEGG pathway enrichment analysis of modules. (**E**) Crosstalk between gene function and signaling pathway; the green circles represent genes, the purple circles represent GO terms and the green boxes represent KEGG terms. (**F**) Interactions among the four modules. (**G**) Ten hub genes are identified.

**Figure 6 genes-11-01187-f006:**
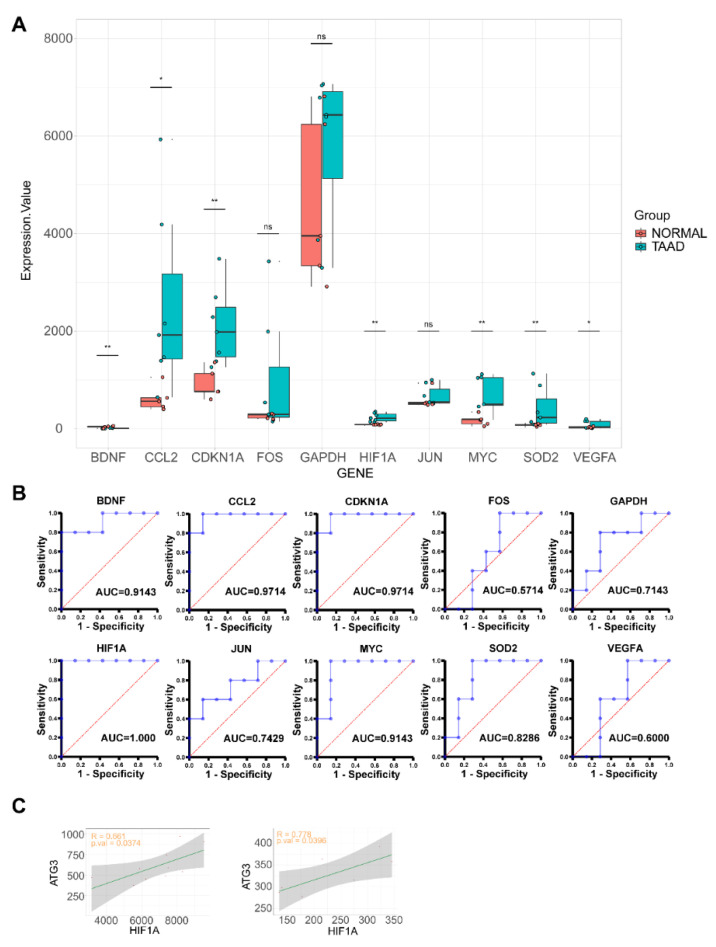
Hub gene validation and receiver operating characteristic (ROC) analysis. (**A**) Hub gene expression value in GSE52093; (**B**) ROC analysis of hub genes in GSE52093. (**C**) Correlation between *HIF1A* and *ATG3*; left is in our dataset, right is in GSE52093.

**Table 1 genes-11-01187-t001:** Clinical information of TAADs and normal samples.

	TAAD (*n* = 10)	NORMAL (*n* = 10)	*p*-Value
Age (years)	59.3 ± 3.9	60.9 ± 3.0	0.75
Male (%)	5 (50%)	4 (40%)	1.00
Height (cm)	168 ± 1.5	163 ± 2.4	0.07
Weight (kg)	71.3 ± 4.5	63.3 ± 3.0	0.16
BMI (kg/m^2^)	25.0 ± 1.4	23.7 ± 0.8	0.45
Aortic diameters (mm)	55.7 ± 9.0	ND	--
Smoking	0 (0%)	0 (0%)	1.00
Hypertension	7 (70%)	3 (30%)	0.18
Diabetes	1 (10%)	0 (0%)	1.00
Alcoholism	0 (0%)	0 (0%)	1.00
CKD	0 (0%)	0 (0%)	1.00
Stroke	0 (0%)	0 (0%)	1.00

BMI, body mass index; ND, not detected.

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
