# Peer review of "Exaggerated Autophagy in Stanford Type A Aortic Dissection: A Transcriptome Pilot Analysis of Human Ascending Aortic Tissues"

_genes, 2020, doi:10.3390/genes11101187_

Round 1

Reviewer 1 Report

The authors presented a very thorough and well written manuscript surrounding the topic of Stanford type A aortic dissection (TAAD), using human samples to identify and characterize molecules that may potentially be significant role players in this acute aortic syndrome. The authors identified several candidates and set the stage for future studies, laying the foundation for more sophisticated experiments. There are only a few minor suggestions:

1) Font: Where possible please increase the font of all figure.

2) Size: Particularly for the various webs in Figure 5. Please provide a larger version of these webs that can be further analyzed in the supplemental text. 

Author Response

The authors presented a very thorough and well written manuscript surrounding the topic of Stanford type A aortic dissection (TAAD), using human samples to identify and characterize molecules that may potentially be significant role players in this acute aortic syndrome. The authors identified several candidates and set the stage for future studies, laying the foundation for more sophisticated experiments. There are only a few minor suggestions:

1) Font: Where possible please increase the font of all figure.

2) Size: Particularly for the various webs in Figure 5. Please provide a larger version of these webs that can be further analyzed in the supplemental text.

Response: We thank the reviewer very much for the suggestions. We made following changes according to the suggestions:

1) We have increased the label fonts of all figures and replace the old version in the manuscript.

2) We provide a larger version of 3 panels of Figure 5 in the supplementary files (Figure S2-S4).

Reviewer 2 Report

The authors report a transcriptomic analysis of ascending aortic tissue in 20 ascending aortic tissues, 10 with TAAD and 10 described as "normal".

Although this may be a Journal style, I'm not used to seeing results in the Introduction "Through the preliminary analysis of the whole transcriptomics data, we found that the significant activation of inflammation and cell death and the obvious degenerative changes of smooth muscle cells, which all indicate that autophagy plays an important role in TAAD."

It is known that there differences in transcriptional expression both longitudinally and circumferentially in the human aorta, much of which can be explained by embryologic origin and perhaps hemodynamics. Please describe exactly which portion of the aorta was sampled for each group.

Table 1. Please provide the measure root and ascending aortic diameters for both groups.

Aneurysmal, but non-dissected aortas, have been observed to have large transcriptional differences compared with "normal" aorta. On the other hand, many dissections occur at relatively low aortic diameters. It would have been highly advantageous to have included dilated and aneurysmal aortas in this analysis.

Did you exclude patients with syndromic disease eg: Marfan, Loeys-Dietz, or those with other known aneurysm-associated variants. Some sequencing would take care of this issue.

The use of whole aortic samples yields intrinsic differences in transcriptional profile across the layers of the aorta. It would have been advantageous to laser-dissect these layers and describe transcription in each layer. Alternatively, single cell sequencing would have been a pinnacle methodology describing differences between cell-types, but obviously, very expensive.

Page 2. "The Benjamini and Hochberg corrected P value of < 0.05 and | LogFC | > 1 were defined as the selection threshold for selecting the DEGs." Although these cut points have been used by prior authors, I think there is a very rational argument to be made for using stricter cut points. Are your results substantially changed by this change?

Page 4. "Volcanic maps" are usually called volcano plots. Please identify the autophagy genes on the volcano plot. Please provide the fold-change and P value for two lists in an on-line supplement: (i) all of the 847 autophagy genes; (2) all the 1525 differentially expressed genes and mark the 275 genes listed in Figure 4.

I have mixed feelings about confirmation of transcriptional findings by RT-PCR. This might have added some rigor to identification of differentially-expressed genes.

Other points:
There are a few grammatical and language errors.

Please increase the font size used for labels in the figures. I can't read many of them.

Author Response

The authors report a transcriptomic analysis of ascending aortic tissue in 20 ascending aortic tissues, 10 with TAAD and 10 described as "normal".

1) Although this may be a Journal style, I'm not used to seeing results in the Introduction "Through the preliminary analysis of the whole transcriptomics data, we found that the significant activation of inflammation and cell death and the obvious degenerative changes of smooth muscle cells, which all indicate that autophagy plays an important role in TAAD."

Response: As the reviewer suggested, we have deleted “These results implicate that exaggerated autophagy participates in the pathological process of aortic dissection and may provide us new insight for further basic research of TAAD” in the Introduction part (Page 2, Line 46).

2) It is known that there differences in transcriptional expression both longitudinally and circumferentially in the human aorta, much of which can be explained by embryologic origin and perhaps hemodynamics. Please describe exactly which portion of the aorta was sampled for each group.

Response: We really thank the reviewer for the helpful suggestion. We have detailed the description of which portion of the aorta was sampled for each group and put this information in the Materials and Methods part (Page 2, Line 75): The TAAD ascending aorta sample was cut above the sinutubular junction and completely transected just proximal to the origin of the brachiocephalic artery, whereas the normal ascending aorta sample was collected by aortic punch during CABG.

3) Table 1. Please provide the measure root and ascending aortic diameters for both groups.

Response: As the reviewer requested, we added the ascending aortic diameters data, which measured based on the results of the preoperative Computed Tomography Angiography (CTA) examination for TAAD group in Table 1 (Page 4, Line 128). However, we did not record the aortic diameters for normal group because routine inspections of these CAD patients did not report aorta abnormalities.

4) Did you exclude patients with syndromic disease eg: Marfan, Loeys-Dietz, or those with other known aneurysm-associated variants. Some sequencing would take care of this issue.

Response: We have excluded patients with hereditary diseases such as Marfan syndrome, Loeys-Dietz syndrome and bicuspid aortic valve malformation. We added this statement in the Materials and Methods part (Page 2, Line 66).

5) The use of whole aortic samples yields intrinsic differences in transcriptional profile across the layers of the aorta. It would have been advantageous to laser-dissect these layers and describe transcription in each layer. Alternatively, single cell sequencing would have been a pinnacle methodology describing differences between cell-types, but obviously, very expensive.

Response: We agree with the reviewer that intrinsic differences could not be avoided in transcriptional profile across the layers of the aorta with use of whole aortic samples. We really thank the reviewer for the helpful suggestion, and we would like to fully consider dissecting these layers or single-cell sequencing analysis in the future studies.

6) Page 2. "The Benjamini and Hochberg corrected P value of < 0.05 and | LogFC | > 1 were defined as the selection threshold for selecting the DEGs." Although these cut points have been used by prior authors, I think there is a very rational argument to be made for using stricter cut points. Are your results substantially changed by this change?

Response: To address the Review’s concern, we used stricter cut points (adjusted P value <0.01 and | LogFC | >1) for selecting the DEGs. With the new threshold we identified 441 upregulated genes were and 784 downregulated genes, in which 64 autophagy-related genes were included. Then we performed GO and KEGG analyses based on the new DEGs, and found that the enrichment items hardly changed.

7) Page 4. "Volcanic maps" are usually called volcano plots. Please identify the autophagy genes on the volcano plot. Please provide the fold-change and P value for two lists in an on-line supplement: (i) all of the 847 autophagy genes; (2) all the 1525 differentially expressed genes and mark the 275 genes listed in Figure 4.

Response: As the Reviewer suggested, we have changed "Volcanic maps" to “Volcano plot” in Figure 2 legend (Page 5, Line 135), and identified the autophagy genes on a new volcano plot (In order to make the picture clearer, we removed the not-significant dots in the original plot) and put it in the supplementary files (Figure S1). We also provide two excel files in the supplements containing the lists of 847 autophagy genes and all the 1525 differentially expressed genes with fold-change and P value, and the genes listed in Figure 4 were highlighted in the excel file.

8) I have mixed feelings about confirmation of transcriptional findings by RT-PCR. This might have added some rigor to identification of differentially-expressed genes.

Response: To address the Reviewer’s concern, we validated the hub genes expression by qPCR in ascending aortic tissues from other 4 “normal” and 7 TAAD patients. Results showed that SOD2, HIF1A and VEGFA were also significantly increased in TAAD ascending aortic tissues (Page 3, Line 112; Page 9, Line 197), which were consistent with the validation results by use of dataset GSE52093 downloaded from Gene Expression Omnibus database. We have put the qPCR results in Supplementary Figure S5.

9) Please increase the font size used for labels in the figures. I can't read many of them.

Response: We are sorry for the small font size, and we have increased the label fonts of all figures and replace the old version in the manuscript.

Reviewer 3 Report

Congratulations on a great job!!! 1. Remember that AAD is not a homogeneous group, it is also associated with the bicuspid aortic valve or collagenopathies.
In text there is no defined if AAD group was homogenous.
2. The aortic structure itself appears to be more complex. Apart from lysis, we can deal with disturbances in the structure and proportion of collagen, elastin, overactivity of metalloproteinases,
disturbances in the matrix structure, etc.

2. In the future, I suggest extending the study to take into account the homogenous or hetrogeneity of the groups of patients with aortic dissection

I suggest title change:

Exaggerated autophagy in Stanford type A aortic dissection: A transcriptome pilot analysis of human ascending aortic tissues

I suggest Limitation chapter explication.

I suggest conclusions adequate to the results of the study - that is, the identification of gene differences in the normal and dissected aorta that may be responsible for structural abnormalities predisposing to AAD.

Author Response

Congratulations on a great job!!!

1) Remember that AAD is not a homogeneous group, it is also associated with the bicuspid aortic valve or collagenopathies. In text there is no defined if AAD group was homogenous. 2) The aortic structure itself appears to be more complex. Apart from lysis, we can deal with disturbances in the structure and proportion of collagen, elastin, overactivity of metalloproteinases, disturbances in the matrix structure, etc.

Reponse: We are encouraged by the words of Congratulations on a great job. As the reviewer suggested, we have added details about TAAD patients in the Materials and Methods part (Page 2, Line 66): All TAAD patients were homogenous and none of them have hereditary diseases such as Marfan syndrome, Loeys-Dietz syndrome and bicuspid aortic valve malformation. We will focus on the other mechanisms except of the lysis in our future study.

  1. In the future, I suggest extending the study to take into account the homogenous or hetrogeneity of the groups of patients with aortic dissection. I suggest title change:

Exaggerated autophagy in Stanford type A aortic dissection: A transcriptome pilot analysis of human ascending aortic tissues. I suggest Limitation chapter explication. I suggest conclusions adequate to the results of the study - that is, the identification of gene differences in the normal and dissected aorta that may be responsible for structural abnormalities predisposing to AAD.

Response: We thank the Reviewer for this helpful suggestion. We will collect patients with hereditary aortic diseases for further studies. As the Reviewer suggested, we have changed the title of our manuscript to “Exaggerated autophagy in Stanford type A aortic dissection: A transcriptome pilot analysis of human ascending aortic tissues”. As the Reviewer suggested, we have added explications about the limitation of this study in the Discussion part (Page 11, Line 272): Owing to the urgency to provide treatment to patients with TAAD and its surgical requirements, basic and follow-up patient data were incomplete. This makes it impossible to perform further analysis on molecular features with specific clinical symptoms and postoperative prognosis. Meanwhile, results of this study could not sufficiently illustrate that the aortic dissection is caused by exaggerate autophagy. In order to prove this hypothesis, animal models are still needed for further verification. As the Reviewer suggested, we have modified the conclusions and made them more adequate to our study (Page 12, Line 277): we identified of differentially expressed genes in the normal and dissected ascending aorta that may be responsible for structural abnormalities predisposing to TAAD. We also identified 10 hub genes and the HIF1A-ATG3 axis involved, which could provide new insights into the understanding of aortic dissection.

Round 2

Reviewer 2 Report

Thank you for the changes